# LGViT: A Local and Global Vision Transformer with Dynamic Contextual Position Bias Using Overlapping Windows

Qian Zhou [1] , Hua Zou [1,*] and Huanhuan Wu [2]

1   School of Computer Science, Wuhan University, Wuhan 430072, China
2   College of Information Engineering, Tarim University, Alar 843300, China
*   Correspondence: zouhua@whu.edu.cn

**Abstract:** Vision Transformers (ViTs) have shown their superiority in various visual tasks for the capability of self-attention mechanisms to model long-range dependencies. Some recent works try to reduce the high cost of vision transformers by limiting the self-attention module in a local window. As a price, the adopted window-based self-attention also reduces the ability to capture the long-range dependencies compared with the original self-attention in transformers. In this paper, we propose a Local and Global Vision Transformer (LGViT) that incorporates overlapping windows and multi-scale dilated pooling to robust the self-attention locally and globally. Our proposed self-attention mechanism is composed of a local self-attention module (LSA) and a global self-attention module (GSA), which are performed on overlapping windows partitioned from the input image. In LSA, the key and value sets are expanded by the surroundings of windows to increase the receptive field. For GSA, the key and value sets are expanded by multi-scale dilated pooling to promote global interactions. Moreover, a dynamic contextual positional encoding module is exploited to add positional information more efficiently and flexibly. We conduct extensive experiments on various visual tasks and the experimental results strongly demonstrate the outperformance of our proposed LGViT to state-of-the-art approaches.

**Keywords:** vision transformer; visual backbone; overlapping windows





## 1. Introduction

Convolution Neural Networks (CNNs) [1] have made great progress in computer vision. Encouraged by the revolutionary performance of AlexNet [2] and ResNet [3], a lot of CNNs have been proposed for various visual tasks. Generally, CNNs perform convolution operations with sliding windows, resulting in transitional invariance and locality. The locality inductive bias may limit the receptive fields of CNNs, making them difficult to model the important long-range dependencies. Although the stacked pooling layer and convolution layer in deep CNNs can enlarge the receptive field, the global contextual interactions are still insufficient.

Recently, inspired by the enormous success of transformers in natural language processing (NLP), researchers have tried to apply Transformer [4] in computer vision to achieve global interactions in feature maps. Since Vision Transformer (ViT) [5] demonstrated the amazing potentiality of vision transformers, transformers have achieved similar or better performance than CNNs on image classification [5–10], object detection [11–15] and semantic segmentation [16–20]. As the key design in transformers, the self-attention mechanism enables the model to learn short-range and long-range visual dependencies. However, the full self-attention mechanism on all patches brings about quadratic memory and computational cost with respect to the number of patches, restricting the application in many visual tasks in which dense prediction at pixel-level and high-resolution input images are required.

To solve the above issue, an alternative way is to carry out self-attention on windows or groups so that cost can be reduced into linear complexity [21–23]. Most window-based

methods partition the input images into non-overlapping windows, as can be seen in Figure 1, the computational area of self-attention is limited to a fixed-size window. The number of patches in each window is much smaller than the size of the whole feature map, thus leading to linear complexity and locality. In such a situation, the receptive field of self-attention is restricted in each local window or group, which cripples the modeling power of the original self-attention mechanism. To build connections across windows, halo [22] and shift [23] operations are adopted to exchange information through nearby windows. However, the receptive field is still limited in several neighboring windows at one stage and stacked blocks are needed to achieve more global self-attention like CNNs. For some dense prediction tasks, the global semantic information is of vital importance, therefore, the larger receptive field may lead to better performance.

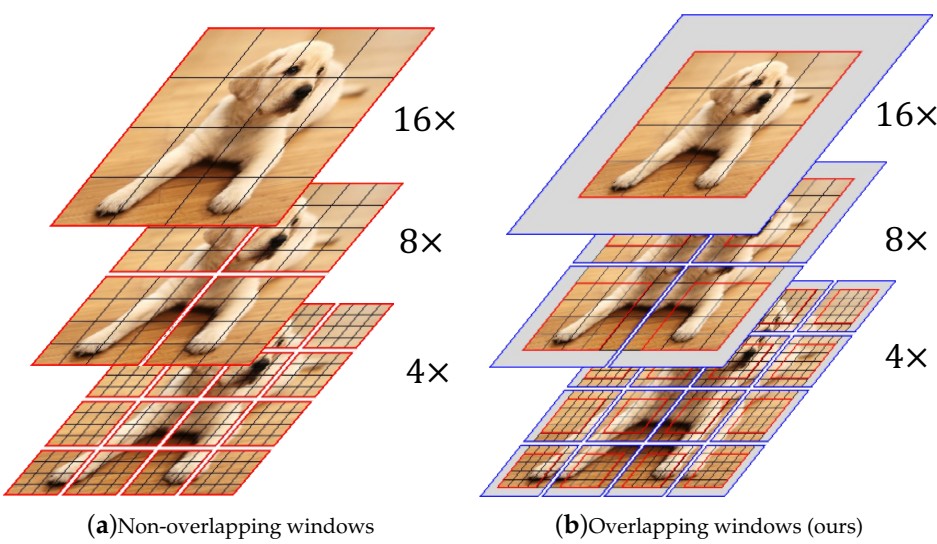

(**a**)Non-overlapping windows　　　　　　　　(**b**)Overlapping windows (ours)

**Figure 1.** Different window partitions used in Swin-Transformer and our proposed LGViT. (**a**) For non-overlapping windows used in Swin-Transformer, the self-attention module is performed within each local limited window. (**b**) For overlapping windows used in LGViT, the feature map will be padded if needed and the self-attention module is conducted within each locally enhanced window.

To overcome the locality limitation of CNNs and reduce the computation cost of transformers, in this paper, we present a novel Local and Global Vision Transformer named LGViT. As illustrated in Figure 2, LGViT is based on our well-designed window-based self-attention mechanism. Such a self-attention mechanism contains a local self-attention module (LSA) to enlarge the receptive field of each window and a global self-attention module (GSA) to obtain contextual information globally. For the naive window-based self-attention mechanism, the receptive field and computational cost are both proportional to the square of window size. In LSA, we adopt the overlapping window design. Specifically, instead of directly enlarging the local window size, we use the overlapping window to expand the key and value sets while keeping the query set in a non-overlapping window. In such a design, the receptive field can be enlarged with a much less cost compared to the naive window-based self-attention mechanism. For the GSA, similar to dilated convolution we utilize the multi-scale dilated pooling with different dilation rates to extend the key set and value set. As a result, global interactions on the whole input feature map/image can be captured.

Like previous Vision Transformers, LGViT also requires position embedding to retain the spatial position information among input patches. A common position embedding is Relative position encoding (RPE) [24], which is independent of the input features and can only be applied when the input size is fixed. Instead, to make it more effective and flexible, we design a dynamic contextual positional encoding module (DCPE) in which the encoding will be changed with the input feature maps. The DCPE takes two relative coordinates

and the current query as input to compute the corresponding position encodings. The DCPE module can be integrated into the transformer blocks with an ignorable cost while making the position encodings applicable to arbitrary input sizes instead of fixed input sizes in RPE.

We shall emphasize that our proposed LGViT is a universal architecture and can be applied to various visual tasks. For simplicity, we conduct experiments on three visual tasks (i.e., image classification, object detection, and semantic segmentation) on three benchmark datasets and achieve very promising results.

In short, the main contributions of our model are listed as follows:

- We propose a novel Local and Global Vision Transformer (LGViT) with a well-designed window-based self-attention mechanism. It contains a local self-attention module (LSA) based on overlapping windows to promote local interactions and a global self-attention module (GSA) with multi-scaled dilated pooling to obtain global contextual information.
- We also design a dynamic contextual positional encoding module (DCPE) to make the relative position embedding more flexible and effective, i.e., applying to variable input size and changing with the input queries.
- Extensive experiments strongly demonstrate that our proposed LGViT achieves outperformance on various visual tasks to state-of-the-art approaches.

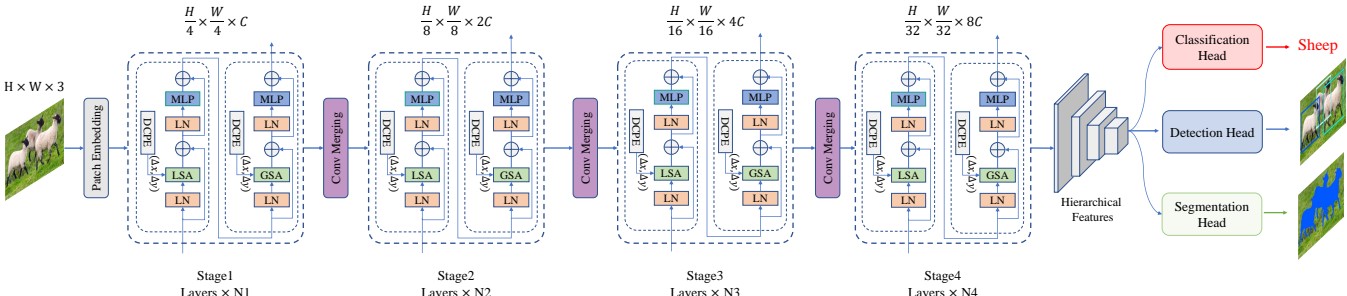

**Figure 2.** The overall architecture of our proposed LGViT, which can be applied to extract hierarchical feature representation for various visual tasks through four stages. Note at each stage, the inner structure is composed of two successive LGViT Blocks, and the LSA and GSA modules are designed to handle both short and long-range dependencies to extract better contextual information. LN means layer normalization.

## 2. Related Work

The related work can be divided into three categories: Vision Transformers, Self-attention Mechanisms, and Position Encoding.

### 2.1. Vision Transformers

CNNs are the primary backbone architectures for computer vision applications, and many tricks like early stopping, gradient clipping, adaptive learning rates, and data augmentation [25] can be used to improve performance. Recently, the tremendous successes of transformers [4] in NLP motivate researchers to explore how to apply transformers for visual tasks. ViT [5] was the first pure transformer architecture in computer vision and achieved state-of-the-art performance on image classification when the model had been trained on a very large dataset. Data-efficient image Transformers (DeiT) [6] further improved ViT [5] by data-efficient training and knowledge distillation, which did not require a very large amount of training data. In addition to image classification, many efforts have been done to make transformers applicable in other visual tasks, including object detection [11–15], semantic segmentation [16–20], image enhancement [26–31] and accurate prediction [32]. Such works have shown the enormous potential of transformers in computer vision. Instead of focusing on a single task, some recent works [8,21,23,33] try to construct a universal backbone based on transformers for various visual tasks. These

backbones take advantage of hierarchical feature maps and can be considered alternatives to CNN backbones like VGG [34] and ResNet [3]. As a variant of ViT, our LGViT is proposed with a well-designed self-attention mechanism to further improve state-of-the-art performances on various visual tasks. LGViT follows a hierarchical design like the recently proposed transformers like Swin Transformer [23].

## 2.2. Self-Attention Mechanisms

As the key design in transformers, the original self-attention is implemented by matrix multiplication as $softmax(\frac{(XW_Q)(XW_K)^T}{\sqrt{d_z}})(XW_V)$, where the projections $W_Q, W_K$, $W_V \in R^{dx \cdot dz}$ are parameter matrices, $d_x$ is the number of queries (keys/values) and $d_z$ is the projection dimension. From this formula, we can see that the computational complexity is quadratic to the sequence length ($\Omega(n^2)$), i.e., input image size in vision transformers. For image classification, the input size is usually not very large, thus the computation cost is acceptable. But for some dense prediction tasks (e.g., object detection and segmentation), the input image size is much larger than image classification. In such a case, the quadratic cost is unbearable. There are two popular ways to overcome this. One is to apply the fine local self-attention [8,21–23], and the other is to adopt sparse global self-attention [7,9,33] to approximate the original self-attention. Liu et al. [23] proposed an efficient window-based self-attention called Swin Transformer, which employs a shifted-window partitioning approach to add the interactions across different local windows. Wang et al. [33] proposed a pyramid vision transformer (PVT) for dense prediction tasks, in which spatial reduction attention (SRA) is presented to approximate the multi-head self-attention in ViT [5]. However, the small-scale information is sacrificed in PVT and the long-range dependencies are limited in a fixed-size window in Swin Transformer. Unlike the existing methods, in order to break the limitations of the fixed-size window, we propose a well-designed window-based self-attention mechanism. More specifically, our proposed self-attention mechanism can be decoupled into LSA and GSA. The LSA is used to enhance the local receptive file and the GSA uses multi-scaled dilated pooling to promote global contextual information.

## 2.3. Position Encoding

Transformers are invariant to the order of the input sequence, which means disrupting the order does not affect the outputs of transformers. As a result, the positional information is ignored in transformers. Thus, position encodings are necessary for transformers to add important spatial information back. The most commonly used methods of position encodings are absolute positional encoding (APE) [4,5,35,36] and relative positional encoding (RPE) [21,23,24,37]. The absolute positional encoding can be computed by sine and cosine functions with different frequencies or predefined by the learnable parameters. The relative position encoding takes the relative coordinates of queries and keys as input and is often implemented by predefined trainable parameters. Both APE and RPE are independent of the input feature maps and are usually designed for a fixed input size. Different from existing APE and RPE, we meticulously design a dynamical contextual positional encoding (DCPE). DCPE uses a lightweight network to generate the position encodings, thus it can be applied to any input resolution. In addition, a global average pooling is utilized to incorporate contextual information.

## 3. Proposed Method

### 3.1. Overall Architecture

The overall architecture of LGViT is illustrated in Figure 2. Like most general backbones [8,21,23] for visual tasks, we also employ a multi-stage design in our model to obtain hierarchical representations. For an input RGB image with a shape of $H \times W \times 3$, we first take a patch embedding layer (i.e., a $4 \times 4$ convolution with stride = 4) to produce $\frac{H}{4} \times \frac{W}{4}$ patch tokens with dimension C. There are four stages in our model. The convolution patch merging layer (i.e., a $3 \times 3$ convolution with stride = 2) is utilized between two adjacent

stages to reduce the number of tokens and double the channel dimension. For the *i*-th stage ($i \in \{1, 2, 3, 4\}$), it consists of a patch embedding/merging layer and $N_i$ LGViT blocks, the shape of output feature maps is $\frac{H}{2^{i+1}} \times \frac{W}{2^{i+1}} \times 2^{i-1}C$, remaining the same as some typical CNNs, e.g., VGG [34] and ResNet [3].

At each stage, the number of LGViT blocks is even. Each LGViT block performs either local self-attention (LSA) or global self-attention (GSA). Note that the local self-attention and the global self-attention are executed by turns. After that, the output multi-scale feature maps are fed into a specific post-processing network to apply in various visual tasks. Take image classification as an example, only the output from the last stage will be pooled and sent into a linear layer to get the predicted class label. As for object detection and segmentation tasks, the feature maps from all stages are fed to the specialized detector head or segmentation head.

### 3.2. LGViT Blocks

The LGViT block takes a similar architecture as Swin Transformer [23], in which the standard multi-head self-attention is replaced by a shifted window-based self-attention. As can be seen in Figure 2, each LGViT block consists of a local self-attention (LSA) module or a global self-attention (GSA) module and a feed-forward network (i.e., the MLP layer). In particular, the dynamic contextual positional encoding module (DCPE) is conducted in both LSA and GSA, making the essential positional information independent of the size of input feature maps but related to the contextual information.

### 3.2.1. Local Self-Attention (LSA) with Overlapping Windows

For most window-based self-attention mechanisms [21,23,38], the input features maps are partitioned into non-overlapping windows, thus limiting the receptive field into fixed window size. Motivated by the great success of CNNs [39–41] in visual tasks, we believe that local information is of great importance in most visual tasks. Therefore, for all query tokens of feature maps, we partition them into non-overlapping windows with the fixed window size (e.g., 7). Meanwhile, for all key and value tokens of feature maps, we partition them into overlapping windows with a larger size (e.g., 13) to promote the local interactions, as seen in Figure 3. The extracted features in the shallow layers of networks contain mainly low-level detail information while the high-level semantic information mainly exists in the deep layers. As a result, the locality inductive bias and larger receptive field are more important in shallow layers. Hence, the expanded size of each overlapping window varies from 3 to 0 according to the layer number, i.e., $Expand = 4 - layer_i, layer_i \in \{1, 2, 3, 4\}$. In summary, in the LSA module, for each query patch in a local window, it will interact with its neighboring patches inside or near the window, which results in a larger receptive field with a lower cost than directly increasing the local window size.

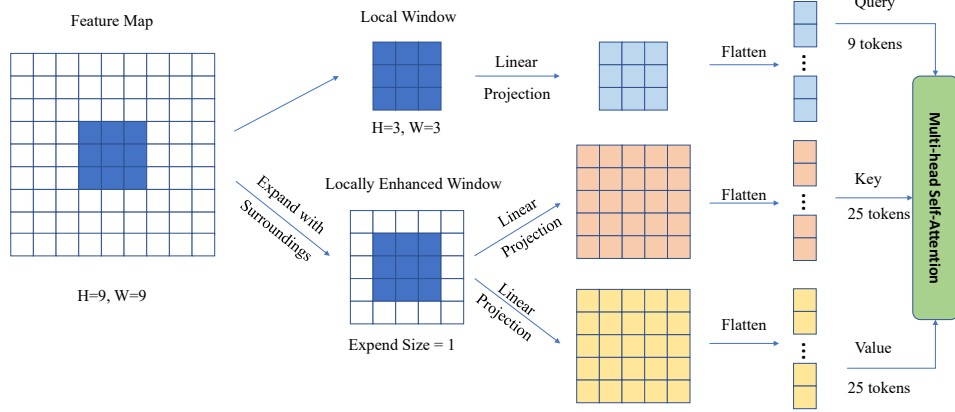

**Figure 3.** An illustration of the LSA module. For each window, the queries are obtained within each non-overlapping window, while the keys and values are based on the overlapping windows.

### 3.2.2. Global Self-Attention (GSA) with Multi-Scale Dilated Pooling

In the LSA module, the interactions of patches are limited mainly in a local window, while the patches located in different windows are not considered enough. To compensate for this, we design the global self-attention module to perform self-attention on the whole feature map, in which a multi-scale dilated pooling with different dilation rates is utilized to expand the key set and value set globally. Specifically, we take a similar way like dilated convolution [42] to implement our pooling operation as shown in Figure 4. The key and value tokens with different dilated rates are firstly aggregated into vectors and then we perform average pooling on them. As a consequence, global interactions among long-distance tokens can be achieved. As a trade-off between the computation cost and performance of self-attention, we set different dilation rates and strides for the pooling operation. For example, to keep the receptive field of pooling consistent with the window size (e.g., 7), the dilation rate will be set as 1 when the kernel size is $7 \times 7$ (large-scale), and the dilation rate will be set as 3 when the kernel size is $3 \times 3$ (small-scale). The sizes of feature maps in the shallow layers are larger than those in the deep layers, thus the stride of the pooling operation varies (e.g., 6,5,4,3) for small-scale pooling while remaining the same (e.g., 7) for the large-scale pooling.

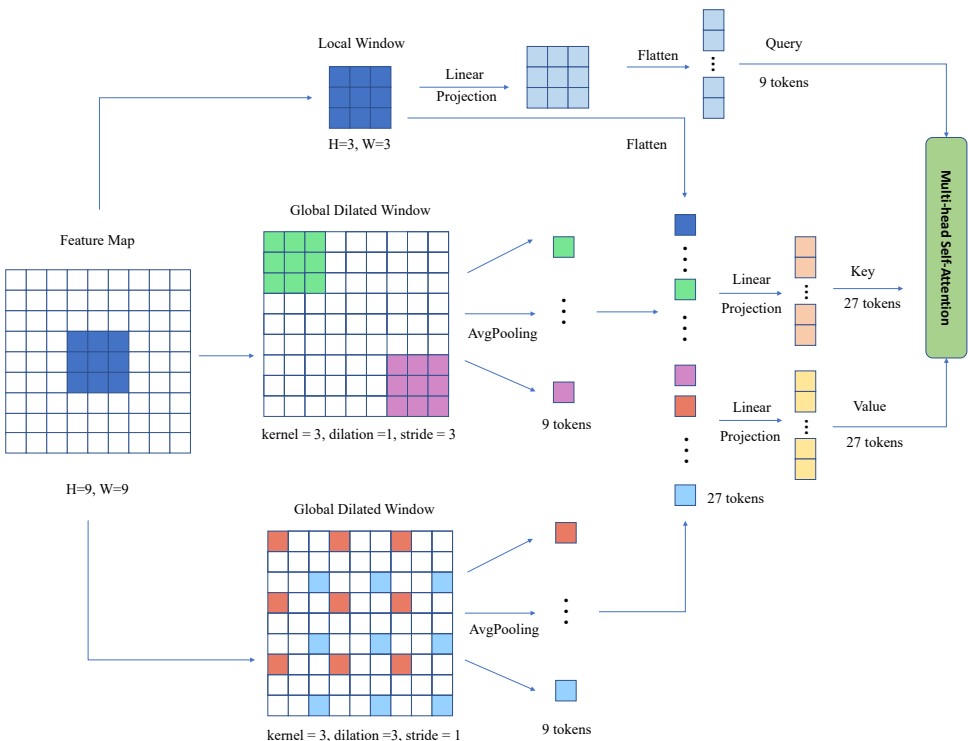

**Figure 4.** An illustration of the GSA module. For each window, the queries, local keys, and local values are obtained within each non-overlapping window, while the global keys and values are computed from multi-scaled dilated pooling.

### 3.3. Contextual Positional Bias

The original self-attention in transformers is equivalent to reordering, which means the output has nothing to do with the way the input tokens are shuffled. This attribute makes the model ignore the important positional information within the 2D images. To retain spatial information, different positional encoding mechanisms have been utilized in existing vision transformers. Among these methods, the relative positional encoding (RPE) [24] has shown its superior performance in vision transformers. Relative positional bias (RPB) [23,43–46] mode is the most commonly used RPE, but can only be applied to the fixed input size. Different from the existing RPB mode, we propose a more general and efficient dynamic contextual positional encoding mode (DCPE). As illustrated in Figure 5,

compared with RPE, the input of DCPE has three parts, i.e., relative coordinate on the height axis, relative coordinate on the width axis, and the query tokens. The output of the DCPE module has the same shape as the attention map obtained by matrix multiplication, thus the output can be directly added to the attention map without any transformation as Equation (1):

$$Attn(Q, K, V) = softmax(\frac{QK^T + AvgPool(Q)R^T}{\sqrt{d_z}}) \tag{1}$$

where the $R^T$ is the relative position bias obtained from the DCPE network. The DCPE network consists of three fully connected layers with layer normalization [47] and ReLU activation [48].

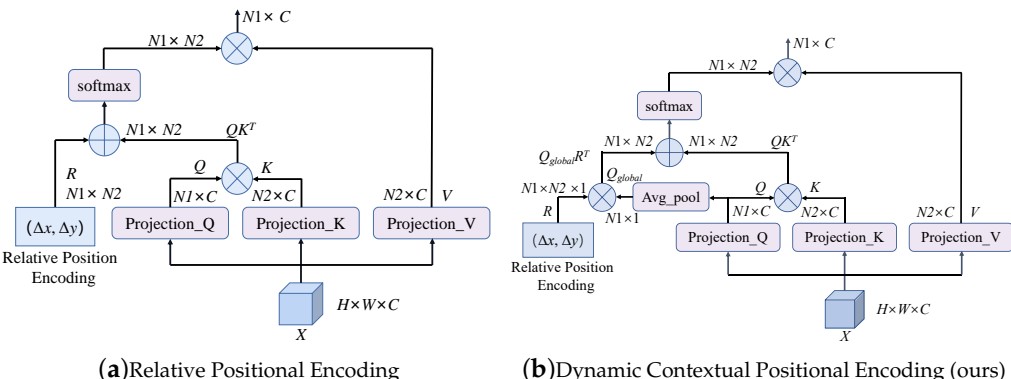

(**a**)Relative Positional Encoding  (**b**)Dynamic Contextual Positional Encoding (ours)

**Figure 5.** Two different positional encoding mechanisms can be used for our proposed LGViT, as well as other Vision Transformers.

Compared with RPE, DCPE can be applied to any input feature size, and the positional encodings will be changed according to the input queries. Therefore, more effective spatial and semantic information will be preserved, leading to a promising performance.

### 3.4. Configurations of LGViT

It is worth noting that compared with the vanilla window-based self-attention proposed in [23], the overlapping window design in our local self-attention (LSA) module and multi-scaled dilated pooling in the global self-attention (GSA) module do not introduce any extra parameters. Thanks to the proposed dynamic contextual position encoding (DCPE) mode, the position encodings can be applied to any number of queries and keys. Hence, the expanded size of overlapping windows in LSA, the strides, dilation rates, and kernel size of pooling in GSA can be set as any values with no need to train from scratch during the test phase. In some other window-based Transformers [23,49], the position encodings are highly related to the number of queries and keys. Thus, the window size in testing must remain the same as the training, restricting the application to various input sizes. In contrast, in the proposed novel window-based self-attention, the window size can be changed according to the input size without re-training. Considering the different computational costs for three typical visual tasks (e.g., image classification, object detection, and semantic segmentation), we conduct three special designs according to the input size, as can be seen in Table 1.

**Table 1.** The three configurations for different visual tasks, i.e., image classification, object detection, and semantic segmentation, on three corresponding benchmark datasets, i.e., Imagenet-1K, COCO2017, and ADE20K, respectively.

| Dataset | Layer Name | Imagenet-1K | COCO2017 | ADE20K |
|---|---|---|---|---|
| **Dataset** | **InputSize** | 224 × 244 | 1280 × 800 | 512 × 512 |
| | **Window Size** | 7 | 8 | 8 |
| Stage1 | Patch Embedding | Conv2d (in = 3, out = 96, kernel = 4, padding = 0, stride = 4) | | |
| | LSA GSA | expand_size = 3 pooling = [(7,1,7), (3, 3, 6)] | expand_size = 4 pooling = [(32,1,32), (16, 2,31)] | expand_size = 4 pooling = [(16,1,16), (8, 2,15)] |
| Stage2 | Conv Merging | Conv2d (in = 96, out = 192, kernel = 3, padding = 1, stride = 2) | | |
| | LSA GSA | expand_size = 2 pooling = [(7,1,7), (3, 3, 5)] | expand_size = 3 pooling = [(8,1,8), (8, 2,15)] | expand_size = 3 pooling = [(8,1,8), (4, 2,7)] |
| Stage3 | Conv Merging | Conv2d (in = 192, out = 384, kernel = 3, padding = 1, stride = 2) | | |
| | LSA GSA | expand_size = 1 pooling = [(7,1,7), (3, 3, 4)] | expand_size = 2 pooling = [(8,1,8), (4, 2,7)] | expand_size = 2 pooling = [(8,1,8), (4, 2,6)] |
| Stage4 | Conv Merging | Conv2d (in = 384, out = 768, kernel = 3, padding = 1, stride = 2) | | |
| | LSA GSA | expand_size = 0 — | expand_size = 1 pooling = [(8,1,8), (4, 2,6)] | expand_size = 1 pooling = [(8,1,8), (4, 2,5)] |

*3.5. Complexity Analysis*

The computation complexity of local-global self-attention in LGViT is analyzed as follows:

For vanilla self-attention [5] with input feature maps $x \in R^{H \times W \times C}$, the computational complexity is shown as Equation (2):

$$\Omega(SA) = 4HWC^2 + 2(HW)^2C \tag{2}$$

where $H, W, C$ are the height, width, and channel dimensions of feature maps, respectively.

A window-based self-attention [23] partitions the feature map into $\frac{H}{M} \times \frac{W}{M}$ non-overlapping windows $x_w \in R^{M \times M \times C}$, and the complexity is Equation (3):

$$\Omega(WSA) = 4HWC^2 + 2M^2HWC \tag{3}$$

where M is the fixed window size.

In our local self-attention (LSA) module, the key set and value set are expanded by overlapping windows. For each window, the total number of queries is $N_1 = M \times M$, while the total number of keys and value are both $N_2 = (M + E)^2$, thus the complexity is Equation (4):

$$\Omega(LSA) = 4HWC^2 + 2N_2HWC \tag{4}$$

where $M$ is the local window size, and $E$ is the expanded size of the current layer.

In the global self-attention module, the key set and value set are expanded by multi-scaled pooling. The total number of keys and values in Equation (4) should be as Equation (5):

$$N_2 = N_1 + \Sigma_{i=1}^{n} O_i^2 \tag{5}$$

where $N_1$ is the number of queries, and $O_i$ means the output size of i-th scaled pooling. A detailed complexity comparison of vision transformers can be seen in Table 2.

**Table 2.** Complexity comparison of different vision transformers. $N = h \times w$, $h$ and $w$ are the height and width of the input image.

| Method | Complexity |
|---|---|
| ViT [5] | $\Omega(N^2)$ |
| DeiT [6] | $\Omega(N^2)$ |
| PvT [50] | $\Omega(N^2)$ |
| CvT [7] | $\Omega(N^2)$ |
| Twins [51] | $\Omega(N)$ |
| Swin-T [23] | $\Omega(N)$ |
| LGViT (ours) | $\Omega(N)$ |

## 4. Experiments

To verify the effectiveness of our architecture as a universal backbone, we conduct experiments on three representative visual tasks (i.e., image classification, object detection, and semantic segmentation). Specifically, the image classification is done on the Imagenet1K [52] dataset, the object detection is conducted on the COCO2017 [53] dataset, while the semantic segmentation is finished on the ADE20K [54] dataset. Finally, we perform comprehensive ablation experiments to show the important design of our backbone.

### 4.1. Image Classification on the ImageNet1K Dataset

**Experiment Settings** There are 1.28 M training images and 50 K validation images from 1000 classes in Imagenet1K [52] dataset, we train our model for 300 epochs with the same training settings as other vision transformers [9,21,23] are employed. We take an AdamW [55] optimizer with a cosine decay learning rate scheduler and 20 epochs of linear warm-up. The input image size is set as $224 \times 224$ by using the same data augmentation and regularization strategies in Swin Transformer [23], including RandAugment [56], Mixup [57], Cutmix [58], random erasing [59] and stochastic depth [60]. The proposed model is trained on 8 RTX 3090 GPUs with a batch size of 1024, an initial learning rate of 0.001, and a weight decay of 0.05. For classification, we use the top-1 accuracy as the metric.

**Results** The results are shown in Table 3. It is obvious that the proposed LGViT outperforms other methods with similar FlOPs and higher accuracy. Specifically, LGViT improves over the newly proposed Swin transformer [23] 0.8% and is 1.3% better than the Deit-Small [6]. The ViL-Small [61] achieves close performance with our model but we still outperform it with 0.1G FLOPs less and 0.1% accuracy higher.

**Table 3.** Performance comparison of image classification task on the ImageNet-1K dataset for different models. All models are trained and evaluated on $224 \times 224$ resolution.

| Method | #Params | FLOPs | Top-1 Acc |
|---|---|---|---|
| ResNet-50 [3] | 25 M | 4.1 G | 76.2 |
| Reg-4G [62] | 21 M | 4.0 G | 80.0 |
| DeiT-S [6] | 22 M | 4.6 G | 79.8 |
| PVT-S [33] | 25 M | 3.8 G | 79.8 |
| T2T-14 [63] | 22 M | 5.2 G | 81.5 |
| ViL-S [61] | 25 M | 4.9 G | 82.0 |
| TNT-S [64] | 24 M | 5.2 G | 81.3 |
| CViT-15 [65] | 27 M | 5.6 G | 81.0 |
| CPVT-S [66] | 23 M | 4.6 G | 81.5 |
| NesT-T [67] | 17 M | 5.8 G | 81.5 |
| CAT-S [68] | 37 M | 5.9 G | 81.8 |
| CvT-13 [7] | 20 M | 4.5 G | 81.6 |
| Swin-T [23] | 29 M | 4.3 G | 81.3 |
| LGViT (ours) | 30 M | 4.8G | **82.1** |

*4.2. Object Detection on the COCO Dataset*

**Experiment Settings** The COCO [53] dataset contains 118 K training and 5 K valida-tion images, the typical RetinaNet [69] in MMDetection [70] is used as the object detection head. Due to the DCPE module in our backbone, our model can be applied to any reso-lution, thus, the model can be initialized with weights pre-trained on ImageNet1K [52]. Following Swin Transformer [23], we use $1\times$ schedule training with 12 epochs. For the $1\times$ schedule, we resize the image's shorter side to 800 while keeping its longer side no more than 1333. We take Adam [55] optimizer with an initial learning rate of $1 \times 10^{-4}$. The batch size is set to 16 on 8 RTX 3090 GPUs. Note that the configuration is a little different from the classification task. For object detection, average precision under different Intersection over Union (IoU) thresholds ($AP^b_{threshold}$) is taken as the metrics. (More details about coco metrics can be found in https://cocodataset.org/#detection-eval, accessed on 1 February 2023)

**Results** The results can be seen in Table 4. Even though we change the configura-tion of object detection compared to image classification, our model still achieves better performance than other models. RegionViT-B's [71] performance is similar to ours, but we are better than them with less parameters and FLOPs. Also, we outperform Swin Transformer [23].

**Table 4.** Performance comparison of object detection task on the COCO2017 dataset for different models with RetinaNet. The FLOPs are computed on $1280 \times 800$ resolution.

| Method | #Params | FLOPs | $AP^b$ | $AP^b_{50}$ | $AP^b_{75}$ |
|---|---|---|---|---|---|
| ResNet-50 [3] | 37.7 M | 234.0 G | 36.3 | 55.3 | 38.6 |
| CAT-B [68] | 62.0 M | 337.0 G | 41.4 | 62.9 | 43.8 |
| ViL-M [61] | 50.8 M | 338.9 G | 42.9 | 64.0 | 45.4 |
| RegionViT-B [71] | 83.4 M | 308.9 G | 43.3 | 65.2 | 46.4 |
| Swin-T [23] | 38.5 M | 245.0 G | 41.5 | 62.1 | 44.2 |
| LGViT (ours) | 40 M | 261 G | **43.4** | **65.3** | **46.9** |

*4.3. Semantic Segmentation on the ADE20K Dataset*

**Experiment Settings** For the semantic segmentation task, ADE20K [54] is the most commonly used dataset, covering a broad range of 150 semantic categories with 20K images for training and 2K images for validation. Similar to models used on the COCO [53] dataset, we initialize the backbone with weights pre-trained on ImageNet1K [52] and take UperNet [72] in MMSegmentation [73] as the segmentation head. For UPerNet [72], an AdamW [55] optimizer with an initial learning rate of $1 \times 10^{-4}$ and a weight decay of 0.01 is used, the models are trained for 160K iterations with a batch size of 16 on 8 RTX 3090 GPUs. In this task, we use mean Intersection over Union (mIoU) as the metric.

**Results** The results are illustrated in Table 5. Like object detection, LGViT shows better performance than others. We achieved similar FLOPs with Swin Transformer [23] even though we expand the key set and value set during the LSA and GSA module. This is because we set the window size in our model as 8 while in Swin [23] the window size is 7, thus the number of windows in our model is less than Swin [23].

**Table 5.** Performance comparison of semantic segmentation task on the ADE20K dataset for different models. The FLOPs are computed on $2048 \times 512$ resolution.

| Method | #Params | FLOPs | mIoU |
|---|---|---|---|
| ResNet-101 [3] | 86 M | 1029 G | 44.9 |
| Shuffle-T [50] | 60 M | 949 G | 46.6 |
| TwinsP-S [51] | 54.6 M | 919 G | 46.2 |
| Twins-S [51] | 54.4 M | 901 G | 46.2 |
| Swin-T [23] | 60 M | 945 G | 44.5 |
| LGViT (ours) | 62 M | 946 G | **47.1** |

### 4.4. Ablation Study

#### 4.4.1. Effectiveness of Overlapping Windows for Key and Value Set in the LSA Module

For most window-based vision transformers [21,23,38], the input images are partitioned into non-overlapping windows so that all query patches within a window share the same key set, which is friendly to memory access on hardware. However, this will limit self-attention to a small local window. To improve local self-attention, we use non-overlapping windows to get queries and overlapping windows to obtain keys and values. In such a situation, the key set and value set for query patches inside a window are still shared while the receptive field can be enlarged with an acceptable cost. The effects of overlapping windows in LSA module can be seen in Table 6. In all visual tasks, the overlapped windows design does improve the performance, especially on object detection (+0.9) and semantic segmentation (+0.7). We attribute it to the essential information for localization, which is mainly related to the local features.

**Table 6.** Effectiveness of the proposed overlapping windows in the LSA module. Metrics on Imagenet1K is top-1 accuracy, on COCO2017 is $AP^b$, and on ADE20K is mIoU.

| Dataset | Overlap? | #Params | FLOPs | Metrics |
|---|---|---|---|---|
| Imagenet1K | $\checkmark$ | 30 M | 4.8 G | 82.1 |
| | $\times$ | 30 M | 4.6 G | 81.7 |
| COCO2017 | $\checkmark$ | 40 M | 261 G | 43.4 |
| | $\times$ | 40 M | 257 G | 42.5 |
| ADE20K | $\checkmark$ | 62 M | 946 G | 47.1 |
| | $\times$ | 62 M | 945 G | 46.4 |

#### 4.4.2. Effectiveness of Multi-Scaled Dilated Pooling in the GSA Module

Even if the design of overlapping windows can enlarge the receptive field of self-attention, self-attention is still limited in a local area near the window. To model the long-range dependencies, we utilize multi-scaled dilated pooling with different dilation rates to expand the key and value set. Due to this design, some patches far from the current window can be represented by a key or a value in self-attention. As shown in Table 7, the results indicate the effectiveness of using multi-scaled dilated pooling to build connections among far patches. We can see that even if local interactions between close patches are more important, the connections among far patches still improve the performance. Especially, the multi-scale design provides an efficient way to build multi-distance dependencies.

**Table 7.** Effectiveness of the proposed multi-scaled (MS) dilated pooling in the GSA module. Metrics on Imagenet1K is top-1 accuracy, on COCO2017 is $AP^b$, and on ADE20K is mIoU.

| Dataset | MS? | #Params | FLOPs | Metrics |
|---|---|---|---|---|
| Imagenet1K | $\checkmark$ | 30 M | 4.8 G | 82.1 |
| | $\times$ | 30 M | 4.6 G | 81.8 |
| COCO2017 | $\checkmark$ | 40 M | 261 G | 43.4 |
| | $\times$ | 40 M | 257 G | 42.9 |
| ADE20K | $\checkmark$ | 63 M | 946 G | 47.1 |
| | $\times$ | 63 M | 943 G | 46.8 |

#### 4.4.3. Dynamic Contextual Positional Encoding

The dynamic contextual positional encoding (DCPE) mode is specially designed for any input resolution. Besides, the proposed DCPE is related to the input queries, thus the positional information is integrated with input features. We compare the parameters, FLOPs, and metrics on three visual tasks of models with relative position bias (RPB) [24]. The results are shown in Table 8. As can be seen, our DCPE module achieves similar and

even better performance than RPB [24] on all tasks. What's more, the DCPE is more flexible, making our model can be applied to any input resolution and configuration.

**Table 8.** Effectiveness of the proposed dynamic contextual positional encoding mode. Metrics on Imagenet1K is top-1 accuracy, on COCO2017 is $AP^b$, and on ADE20K is mIoU.

| Dataset | Method | #Params | FLOPs | Metrics |
|---|---|---|---|---|
| Imagenet1K | DCPE | 30M | 4.8G | 82.1 |
| | RPB | 30M | 4.7G | 82.0 |
| COCO2017 | DCPE | 40M | 261G | 43.4 |
| | RPB | 40M | 261G | 43.2 |
| ADE20K | DCPE | 63M | 946G | 47.1 |
| | RPB | 63M | 946G | 47.0 |

## 5. Discussion

### 5.1. Peformance of LGViT for Different Visual Tasks

For image classification, LGViT achieves a top-1 accuracy of 82.1% on Imagenet1K, which is the best in compared approaches. From Table 3, we can see vision transformers usually outperform CNNs. The typical Resnet-50 [3] only obtains 76.2% top-1 accuracy while most transformer-based models get an accuracy over 81.0%. This is mainly due to the superiority of vision transformers to learn long-range dependencies. CNN is good at extracting high-frequency information, so it works better locally. The transformer is good at extracting low-frequency information, thus working better globally. For these natural images in Imagenet, the scale is mostly limited to a single object and global features are more important. Therefore, transformers can achieve such a good performance than CNNs. Besides, when adopting a hierarchical design [7,23,68], the performance of vision transformers can be further improved compared with naive ViT [5]. The hierarchical design enables the model to learn multi-scale features as well as introduce locality to perform better. Similar experimental results can also be seen in object detection on COCO2017 and semantic segmentation on ADE20kK. LGViT still outperforms other approaches.

### 5.2. Limitations and Future Work

As described in Section 3.4, we adopt different configurations for the object detection and semantic segmentation tasks compared to the image classification task. This is due to the high computational cost to build global connections in the GSA module when the input resolution becomes very large. Therefore, for the proposed LGViT, when applying it on a new resolution, the configuration should be well-designed to achieve a trade-off between cost and performance. The progress to find the best configuration may be a little time-consuming but is of great value to the training phase. A well-designed configuration of the LSA and GSA modules may result in a great performance. Because of the DCPE mode in LGViT, the configurations can be changed at any time, showing high flexibility by comparison with other window-based self-attention methods [23,49].

However, LSA and GSA do introduce extra computational and memory costs, since for each window, the key set and value are extended by its neighbors or the whole image patches. When applying the model to a very high-resolution application may be difficult. For example, the resolution of an ultra wide field fundus fluorescein angiography (UWFFA) can be $3k \times 4k$. The computational cost of transformers for retinal vessel segmentation using UWFFA may be unaffordable. Developing some practical techniques like global information aggregation [74] to reduce the cost would be necessary to make it more applicable in realistic scenarios. Moreover, our work mainly focus on visual tasks, the great potential of Transformer-like architectures for unified modeling between vision and language has not been explored in this work. We hope to investigate this promising direction in the future.

## 6. Conclusions

In this paper, we proposed a novel local and global vision Transformer (LGViT) with a new window-based self-attention mechanism to promote the local interactions of patches in each window and the global interactions of patches between all patches. Moreover, a dynamic contextual positional encoding module is proposed, making the positional embedding applicable to any input size and related to the input features. Qualities of experiments on three typical visual tasks have demonstrated the superiority of our model, which achieves better performance than other vision transformers and CNNs. Finally, the ablation studies further indicate the effectiveness of the LSA, GSA, and DCPE module in our backbone. Our future work includes extending the current LGViT to more real-world visual applications and vision-language multimodal learning.

**Author Contributions:** Conceptualization, Q.Z. and H.Z.; methodology, Q.Z. and H.Z.; software, Q.Z.; validation, Q.Z.; formal analysis, Q.Z. and H.Z.; writing—original draft preparation, Q.Z. and H.Z.; writing—review and editing, H.Z and H.W. All authors have read and agreed to the published version of the manuscript.

**Funding:** This research is partially supported by Bingtuan Science and Technology Program (No. 2022DB005 and 2019BC008) and Science and Technology Project of Tarim University (No. TDZKZD202104).

**Institutional Review Board Statement:** Not applicable.

**Informed Consent Statement:** Not applicable.

**Data Availability Statement:** Publicly available datasets were used in this study. The Imagenet dataset can be found here: https://image-net.org/download.php (accessed on 1 February 2023), the COCO dataset can be found here: https://cocodataset.org/ (accessed on 1 February 2023), the ADE20K dataset is available at: https://groups.csail.mit.edu/vision/datasets/ADE20K/ (accessed on 1 February 2023).

**Acknowledgments:** The authors acknowledge funding from the Bingtuan Science and Technology Program (2022DB005, 2019BC008).

**Conflicts of Interest:** The authors declare no conflict of interest.

## Appendix A. More Architecture Details

*Appendix A.1. Architecture of DCPE*

For the proposed dynamic contextual position encoding mode (DCPE), we take the same network of dynamic position bias (DPB) proposed in [21] to produce relative position bias. As can be seen in Figure A1, the network consists of three fully connected layers with layer normalization and ReLU activation and two linear projection layers to change the dimension of bias. However, our DCPE as Equation (A1) is different from the DPB as Equation (A2) in [21] on the way added to the attention map. Specifically, our DCPE receives two relative coordinates on the height and width axis and the query token as input, making the bias more related to the global contextual information.

$$Attn(Q, K, V) = softmax(\frac{QK^T + AvgPool(Q)R^T}{\sqrt{d_z}}) \tag{A1}$$

$$Attn(Q, K, V) = softmax(\frac{QK^T + R^T}{\sqrt{d_z}}) \tag{A2}$$

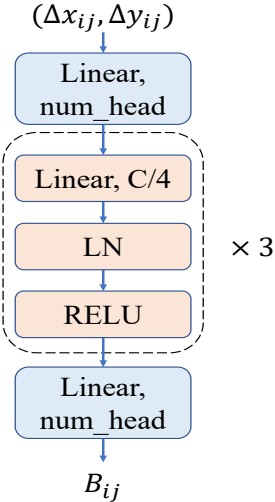

**Figure A1.** An illustration of the network architecture of DCPE in our network, the num_head means the head number in multi-head self-attention, and C represents the embedding dimension of each token.

*Appendix A.2. Importance of LSA and GSA*

The overlapping window partitions in the local self-attention module (LSA) and the multi-scaled dilated pooling operations in the global self-attention module (GSA) are the key designs in our network. To further explore the importance of our proposed LSA and GSA, we conduct two more experiments. One is to use LSA in all LGViT blocks without GSA, and the other is to take GSA in all LGViT blocks without LSA. Quantitative results on three visual tasks are shown in Table A1. We observe that the proposed LSA outperforms GSA for most visual tasks. Specifically, for object detection and semantic segmentation, the performance gaps between LSA and GSA are much larger than image classification. This could be attributed to the dense prediction tasks more rely on low-level localization information which mainly exists in local features.

**Table A1.** Quantitative results of using different self-attention modules. WSA means the original Window-based self-attention proposed in [23].

| Dataset | Modules | #Params | FLOPs | Metrics |
|---------|---------|---------|-------|---------|
| Imagenet1K | LSA&GSA | 30 M | 4.75 G | **82.1** |
| | LSA&LSA | 30 M | 4.76 G | 81.9 |
| | GSA&GSA | 30 M | 4.74 G | 81.8 |
| | LSA&WSA | 30 M | 4.64 G | 81.8 |
| | WSA&GSA | 30 M | 4.63 G | 81.7 |
| COCO2017 | LSA&GSA | 40 M | 261.0 G | **43.4** |
| | LSA&LSA | 40 M | 261.09 G | 43.1 |
| | GSA&GSA | 40 M | 260.91 G | 42.8 |
| | LSA&WSA | 40 M | 256.9 G | 42.9 |
| | WSA&GSA | 40 M | 257.3 G | 42.5 |
| ADE20K | LSA&GSA | 63 M | 946.2 G | **47.1** |
| | LSA&LSA | 63 M | 944.2 G | 46.9 |
| | GSA&GSA | 63 M | 948.2 G | 46.7 |
| | LSA&WSA | 63 M | 943.3 G | 46.8 |
| | WSA&GSA | 63 M | 945.1 G | 46.4 |

## Appendix B. More Visualization Results

Following CBAM [75], we apply the Grad-CAM [76] with different settings to show the heat maps using images from the ImageNet [52] validation set. Using the Grad-CAM,

we can see regions of interest from the results which directly show the effectiveness of the proposed self-attention module. Additionally, we visualize the features from the first two stages like [63] to find out whether the low-level structure features such as edges and lines can be promoted by the LSA, and whether the high-level global features can be well-learned by the GSA. Besides, the effectiveness of DCPE is also visualized.

*Appendix B.1. Visualization of Heat Maps*

As can be seen in Figures A2 and A3, we explore that the proposed LSA usually outperforms GSA when the object is small (the first four columns), however when the number of objects increases (the fifth column) or the size of the object becomes large (the last three columns), the proposed GSA performs better than LSA. Without the proposed DCPE, there will be many prediction errors and the performance is very poor. We argue that GSA tends to large areas but may cause some wrong attention to meaningless regions, while LSA focuses on local patches but may ignore some global information. The position embedding is of great importance to the transformers, when removing it, we get many meaningless attention regions. When combining these three modules, the model can learn both local and global contextual information thus achieving the best performance. This shows the importance and indispensability of the three modules we proposed.

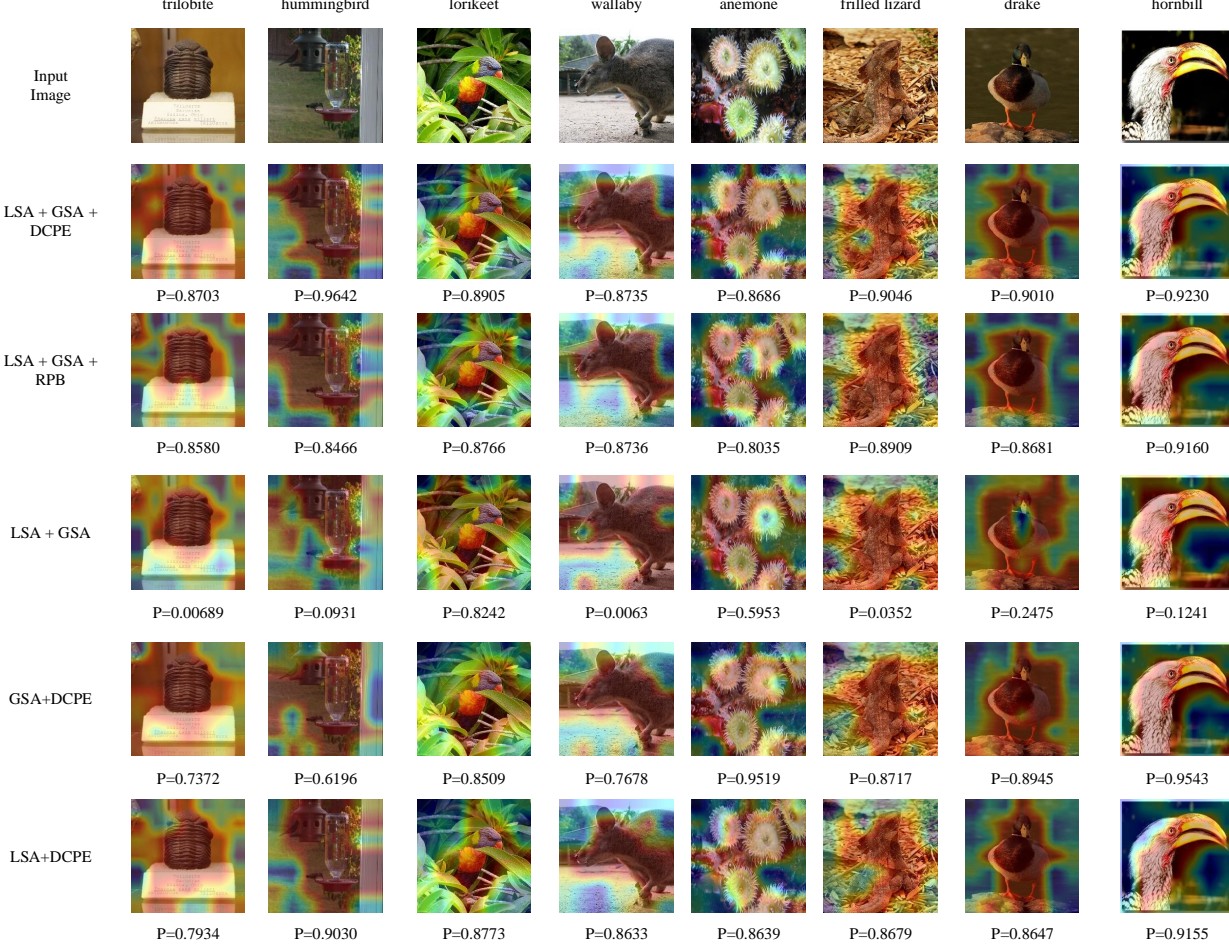

**Figure A2.** Visualization of heat map calculated by Grad-CAM. We compare the visualization results of our network with different self-attention modules and position encoding modes. The grad-CAM visualization is calculated for the last stage outputs. The ground-truth label is shown on the top of each input image and P denotes the softmax score of each network for the ground-truth class.

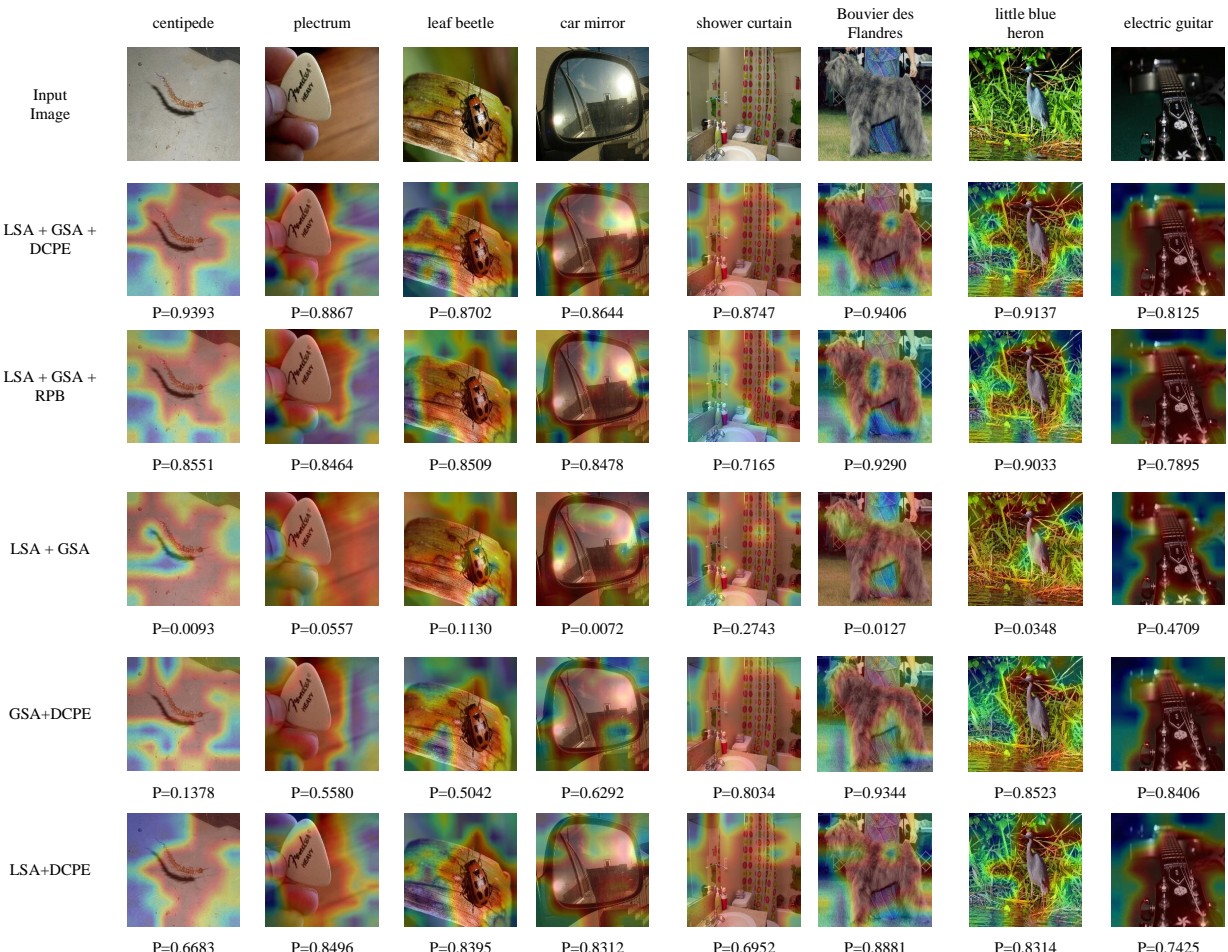

**Figure A3.** Another visualization of heat map calculated by Grad-CAM. As can be seen, when GSA outperforms LSA, the model without position bias performs better (the fifth and the last column). This phenomenon again shows that the location information mainly exists in local features, thus the model with only LSA shows more potential than the model with only GSA on dense prediction tasks as shown in Table 3.

*Appendix B.2. Visualization of Features*

Figures A4 and A5 show the visualization results of features extracted by LGViT and its variants. The feature maps are from the same channel for each column. Without the proposed DCPE, we can see many meaningless noises in the features, especially in Stage 2. The LSA can capture more clear low-level structures like edges and lines while the GSA shows more interest in global semantic information. In consideration of the size of features in four stages (i.e., $56 \times 56$, $28 \times 28$, $14 \times 14$, and $7 \times 7$), we only visualize the first two stages. Note that we do not resize the input images to a larger size (e.g., $1024 \times 1024$) like [63], for this may harm the performance in our experiments. Our LGViT combines the advantages of the LSA, GSA, and DCPE, hence the low-level structures are more clear and the semantic information is fully considered.

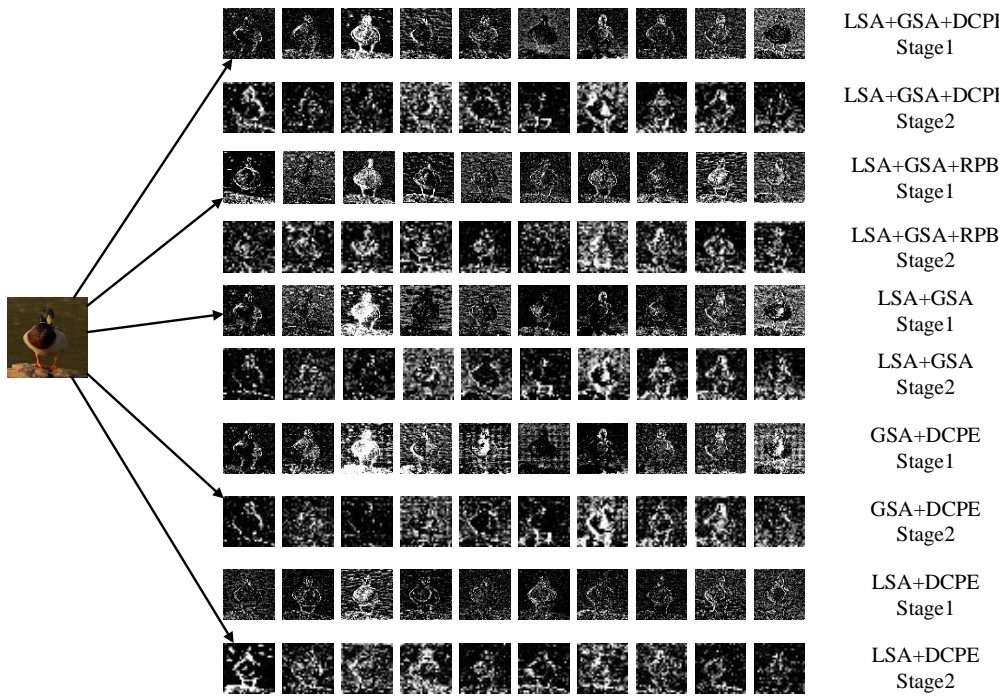

**Figure A4.** Feature visualization of our network with different self-attention modules and position encoding modes. Considering the input size is 224 × 224, we only visualize the outputs of the first two stages.

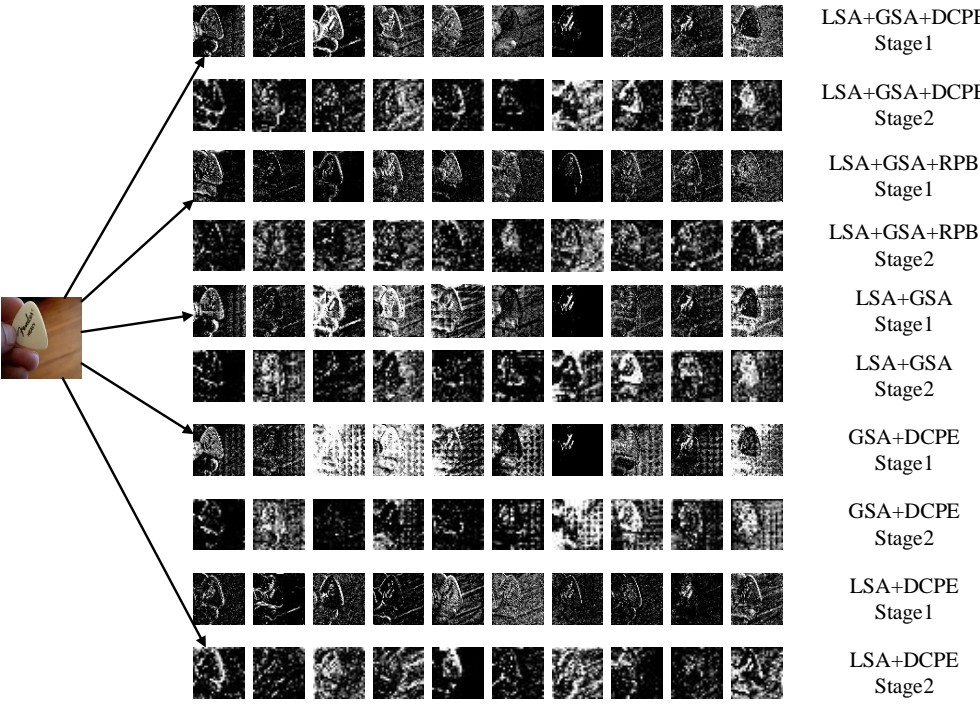

**Figure A5.** Another feature visualization of our network with different self-attention modules and position encoding modes. Only the outputs of the first two stages are visualized.

*Appendix B.3. Comparison of DCPE to RPB*

As shown in Figures A2 and A3, the DCPE used in our model and RPB proposed by [43] achieve similar performance, but the proposed DCPE still outperforms RPB for the considered global contextual information when adding back to transformers. From Figures A4 and A5, we can see that features extracted by DCPE and RPB have similar distribution on space, which is consistent with the similar performances achieved by DCPE and RPB.

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
