# Peer review of "LGViT: A Local and Global Vision Transformer with Dynamic Contextual Position Bias Using Overlapping Windows"

_applsci, doi:10.3390/app13031993_

Round 1

Reviewer 1 Report

* Good paper, need to improve english langauge.

* Why only two different techniques used for encoding mechanisam? 

Author Response

Q1: Good paper, need to improve the English language.

A1: We have carefully checked our paper and improved the writing.

Q2: Why are only two different techniques used for the encoding mechanism?

A2: The most common and popular encodings in Transformers are Relative Position Encoding (RPE) and Absolute Position Encoding (APE). Our proposed Dynamic Contextual Position Encoding (DCPE) improved RPE in flexibility and effectiveness. Specifically, we use a lightweight network (shown in Figure A1) to generate position encodings instead of predefined parameters, thus DCPE can be applied to any input size. In addition, the global average pooling is introduced to integrate contextual information, leading to better performance. In summary, we mainly compared our DCPE to RPE in the experiments since RPE and DCPE usually achieve similar performance in most visual tasks. A more detailed discussion about position encoding can be found in [1].

Reference

[1] Wu K, Peng H, Chen M, et al. Rethinking and improving relative position encoding for vision transformer[C]//Proceedings of the IEEE/CVF International Conference on Computer Vision. 2021: 10033-10041.

Reviewer 2 Report

Please rectify the reviewer comments

Review comments to the author (Applied Sciences -2147184)

In this manuscript, the author wrote an article entitled “LGViT: A Local and Global Vision Transformer with Dynamic Contextual Position Bias Using Overlapping Windows” suitable for publication, but the concerned author has to rectify the below-mentioned minor review comments in the “Applied Sciences”.

 I have made a few observations after reading this work. This article might be accepted after fixing the ensuing minor review comments. However, kindly correct the following:

1.         To what end does the vision transformer serve? In what ways can a vision transformer be utilised in practise?

2.         How many distinct forms of attention mechanisms are there? In what contexts do people typically make use of the attention mechanism?

3.         Is there a correlation between visual acuity and how well you do on a given task? Your feedback on the eye exam's visual acuity should look like what?

4.         What role does visual acuity play in how well you see? How strong must a prescription be for glasses to be considered "light?"

5.         Is it true that transformers have poor results when representing a graph? Is there a particularly challenging transformer?

6.         Are vision transformers resilient to the patch perturbations that occur on github? Are Vision transformers resistant to the effects of the Patchwise perturbation?

7.         Is the use of vision transformers always going to result in a more robust system than the use of convolutional neural networks?

It would be best if you corrected the above comments and resubmitted them according to your expectations.

Author Response

Q1: To what end does the vision transformer serve? In what ways can a vision transformer be utilized in practice?

A1:

About why the vision transformer works: CNN is good at extracting high-frequency information, so it works better locally. The transformer is good at extracting low-frequency information, thus working better globally. In some common datasets like Imagenet, the scale is mostly limited to a single object, and global features are more important. Therefore, transformers can achieve such a good performance.

About how the vision transformer can be utilized: Vision transformers can be applied as the backbone to extract features in most visual tasks. Pure transformer-based architectures can obtain better or similar performance than CNNs. Some recent works try to combine CNNs and Transformers to simultaneously extract global and local features and get good results in medical image segmentation [2,3]. Since Transformers have been the de-facto standard for Natural Language Processing (NLP) tasks and achieved promising results in visual tasks, some works focus on multimodal learning (vision and language) using Transformers [4,5].

Q2: How many distinct forms of attention mechanisms are there? In what contexts do people typically make use of the attention mechanism?

A2: 

About attention mechanisms categories: Attention mechanisms can be divided into two categories: hard attention mechanisms and soft attention mechanisms. Hard attention is usually non-differentiable, the most commonly used is soft attention. Some soft attention focus on spatial attention [6], some works major in channel attention [7], and some try to apply them both [8]. Self-attention can be considered as a variant of soft attention, it reduces the dependence on external information and is better at capturing the internal relevance of data. The self-attention mechanism mainly solves the long-distance dependency problem by computing the interactions between tokens.

About how to use attention mechanisms: The attention mechanism is often used as a plug-and-play module that can be embedded in any part of the network. The attention mechanism makes the network more focused on important and valuable information or features.

Q3: Is there a correlation between visual acuity and how well you do on a given task? Your feedback on the eye exam's visual acuity should look like what?

A3: We consider visual acuity as the visualization results of feature maps as shown in Figure B3 and Figure B4. We can see that the contours (global information) of the object is well preserved in the visualized feature map, showing the effectiveness of our proposed LGViT.

Q4: What role does visual acuity play in how well you see? How strong must a prescription be for glasses to be considered "light?"

A4: We regard visual acuity as the visualization results of feature maps. Due to the LSA module, LGViT is able to capture the local information and features (textures). In addition, LGViT is able to extract global information and features (contours). The visualization results can be seen in the appendix.

Q5: Is it true that transformers have poor results when representing a graph? Is there a particularly challenging transformer?

A5:

About representing a graph: We don't do research about graph neural networks (GNNs), so we don't know much about them. More information about Transformers in graph networks can refer to [9]. Chen et al. [10] think that Transformer naturally overcomes several limitations of GNNs by avoiding their strict structural inductive biases and instead only encoding the graph structure via positional encoding. The authors propose a structure-aware framework that can leverage any existing GNN to extract the subgraph representation, and they show that it systematically improves performance relative to the base GNN model, successfully combining the advantages of GNNs and Transformers.

About challenging transformer: In remote sensing images, the scale of each object is generally small. Local features are more important than global features, so transformers do not have much improvement compared to CNNs. Besides, the high computation cost of the Transformer also limits its application to high-resolution images.

Q5: Are vision transformers resilient to the patch perturbations that occur on github? Are vision transformers resistant to the effects of Patchwise perturbation?

A5: He et al. [11] propose masked autoencoders (MAE) by performing masked image modeling (MIM) using vision transformers. The results shown in [11] are very amazing and promising, even at a masking ratio of 75%, vision transformers can still recover the main objects in the picture. Therefore, I believe that vision transformers are robust to patchwise perturbation. More information about MIM using vision transformers can refer to [12, 13].

Q6: Is the use of vision transformers always going to result in a more robust system than the use of convolutional neural networks?

A6: Recently, many researchers have focused on the robustness between vision transformers and CNNs [14, 15, 16]. Benz et al. [14] evaluate the adversarial robustness of ViTs under several adversarial attack setups and benchmark against CNNs. The authors find that ViT is more robust than their CNN models. Mao et al. [15] conduct systematic evaluations on components of ViTs in terms of their impact on robustness to adversarial examples, common corruptions, and distribution shifts. The authors also propose Robust Vision Transformer (RVT) by using and combining robust components as building blocks of ViTs, which has superior performance with strong robustness. Tang et al. [16] find that transformers are more robust against adversarial noises while less robust against natural noises compared to CNNs under aligned settings. In general, Transformers and CNNs have their own advantages and disadvantages in terms of robustness. Under different situations, their robustness may be different.

References

[1] Wu K, Peng H, Chen M, et al. Rethinking and improving relative position encoding for vision transformer[C]//Proceedings of the IEEE/CVF International Conference on Computer Vision. 2021: 10033-10041.

[2] Chen J, Lu Y, Yu Q, et al. Transunet: Transformers make strong encoders for medical image segmentation[J]. arXiv preprint arXiv:2102.04306, 2021.

[3] Zhang Y, Liu H, Hu Q. Transfuse: Fusing transformers and cnns for medical image segmentation[C]//International Conference on Medical Image Computing and Computer-Assisted Intervention. Springer, Cham, 2021: 14-24.

[4] Radford A, Kim J W, Hallacy C, et al. Learning transferable visual models from natural language supervision[C]//International Conference on Machine Learning. PMLR, 2021: 8748-8763.

[5] Li B, Weinberger K Q, Belongie S, et al. Language-driven Semantic Segmentation[C]//International Conference on Learning Representations. 2021.

[6] Hu J, Shen L, Sun G. Squeeze-and-excitation networks[C]//Proceedings of the IEEE conference on computer vision and pattern recognition. 2018: 7132-7141.

[7] Wang X, Girshick R, Gupta A, et al. Non-local neural networks[C]//Proceedings of the IEEE conference on computer vision and pattern recognition. 2018: 7794-7803.

[8] Woo S, Park J, Lee J Y, et al. Cbam: Convolutional block attention module[C]//Proceedings of the European conference on computer vision (ECCV). 2018: 3-19.

[9] Yun S, Jeong M, Kim R, et al. Graph transformer networks[J]. Advances in neural information processing systems, 2019, 32.

[10] Chen D, O’Bray L, Borgwardt K. Structure-aware transformer for graph representation learning[C]//International Conference on Machine Learning. PMLR, 2022: 3469-3489.

[11] He K, Chen X, Xie S, et al. Masked autoencoders are scalable vision learners[C]//Proceedings of the IEEE/CVF Conference on Computer Vision and Pattern Recognition. 2022: 16000-16009.

[12] Bao H, Dong L, Piao S, et al. BEiT: BERT Pre-Training of Image Transformers[C]//International Conference on Learning Representations. 2021.

[13] Xie Z, Zhang Z, Cao Y, et al. Simmim: A simple framework for masked image modeling[C]//Proceedings of the IEEE/CVF Conference on Computer Vision and Pattern Recognition. 2022: 9653-9663.

[14] Benz P, Ham S, Zhang C, et al. Adversarial robustness comparison of vision transformer and mlp-mixer to cnns[J]. arXiv preprint arXiv:2110.02797, 2021.

[15] Paul S, Chen P Y. Vision transformers are robust learners[C]//Proceedings of the AAAI Conference on Artificial Intelligence. 2022, 36(2): 2071-2081.

[16] Tang S, Gong R, Wang Y, et al. Robustart: Benchmarking robustness on architecture design and training techniques[J]. arXiv preprint arXiv:2109.05211, 2021.

Reviewer 3 Report

The main contribution and originality of the research should be explained in more detail in the introduction section. In particular, it is necessary to highlight which are the critical points and performance limits of the current LGViT and how they are overcome by the proposed LGVit.

What are the performance pros and cons of the three categories of LGViT discussed in section 2)? Authors should enter a discussion on this point.

Figure 3 is not very significant and does not clearly show the architecture of the LSA model. Also, the caption under the figure is too long; in fact, the descriptive aspects of the LSA models must be inserted in the text and not in the caption of the figure.

The same considerations apply to Figure 4.

Do the performances of the different models in Fig. 2 depend on the choice of parameters? Authors have to explain whether Flops and Accuracy of the proposed model depend on the choice of initial learning rate and weight decay values and which optimization approach is used for the choice of these values.

pay particular attention to describing the meaning of all the column names in tables 3-7 (Apb, Apb50, Apb75, mIoU, metrics). Do metrics mean accuracy?

The word "page" appears several times in the text next to figures, equations or tables. It is necessary to eliminate this writing and make the manuscript fully adherent to the template-structure of the journal.

Author Response

Q1: The main contribution and originality of the research should be explained in more detail in the introduction section. In particular, it is necessary to highlight which are the critical points and performance limits of the current LGViT and how they are overcome by the proposed LGVit.

A1: We have revised the introduction to highlight the shortcomings of existing methods and the innovations of the proposed method.

Q2: What are the performance pros and cons of the three categories of LGViT discussed in section 2)? Authors should enter a discussion on this point.

A2: We have revised the related work and added a discussion about the connections between existing works and our work.

Q3: Figure 3 is not very significant and does not clearly show the architecture of the LSA model. Also, the caption under the figure is too long; in fact, the descriptive aspects of the LSA models must be inserted in the text and not in the caption of the figure. The same considerations apply to Figure 4.

A3: We have updated our figures and explained LSA and GSA more clearly.

Q4: Do the performances of the different models in Fig. 2 depend on the choice of parameters? Authors have to explain whether Flops and Accuracy of the proposed model depend on the choice of initial learning rate and weight decay values and which optimization approach is used for the choice of these values.

A4: The performance (Accuracy) is related to the choice of parameters (batch size, initial learning, optimizer, etc.). The FLOPs are independent of the training settings, it only depends on the input image size and the model design. The detailed training settings on three visual tasks are described in Section 4.1-4.3. For hyperparameters, we mainly follow other similar works since hyperparameter search on such “big” datsets is time-consuming.

Q5: Pay particular attention to describing the meaning of all the column names in tables 3-7 (APb, APb50, APb75, mIoU, metrics). Do metrics mean accuracy?

A6: We have added the explanation of these metrics. APb, APb50, APb75 are average precision under different Intersection over Union (IoU) thresholds, more details can be seen on the coco website (https://cocodataset.org/#detection-eval). In tables 3-7, Metrics on Imagenet1K is top-1 accuracy, on COCO2017 is APb, and on ADE20K is mean Intersection over Union (mIoU).

Q7: The word "page" appears several times in the text next to figures, equations or tables. It is necessary to eliminate this writing and make the manuscript fully adherent to the template-structure of the journal.

A7: We have checked the paper and fixed them. We recompile the project through the online overleaf and a local machine, and do not find this “page” error. The corrected pdf file is provided this time. This issue may be due to the journal submission system compiling. If you still have doubts about this, you can directly view the pdf file we provided.

Reviewer 4 Report

The authors proposed a technique called LGViT which incorporates overlapping windows and multi-scale dilated pooling to robust locally and globally the self-attention. The method and results are interesting. I have some comments for further improvement.

Line 18, etc. There are several places where the word “page” is typed by error. Probably a compiling issue. Please fix.

Line 98, DeiT. Please define all the acronyms the first time they appear. 

Line 106. The following article compares VGG and ResNet architectures and can be added here:

https://doi.org/10.3390/s21238083

Also, some other recently developed visual transformer methods with applications can be added in Related Works, such as

https://doi.org/10.3390/rs14061400

Line 194. Format issue. It should be “Figure 4”. Please check and correct similar issues. Like line 227 that should say “Table 1”. Likewise, when you referring to equation, you should say “equation (x)”.

Please add the computational time (in seconds or big O) for different methods in one of the tables. 

The discussion section is very short. Please first state the objective and discuss your results in the light of other similar studies. Furthermore, please include the limitations of the study and give future directions. 

Finally, please carefully proofread the article and fix typos/punctuation/grammar issues.

Thank you for your contribution 

Author Response

Q1: Line 18, etc. There are several places where the word “page” is typed by error. Probably a compiling issue. Please fix.

A1: We have checked the paper and fixed them. We recompile the project through the online overleaf and a local machine, and do not find this “page” error. The corrected pdf file is provided this time. This issue may be due to the journal submission system compiling. If you still have doubts about this, you can directly view the pdf file we provided.

Q2: Line 98, DeiT. Please define all the acronyms the first time they appear.

A2: We have checked the paper and modified them.

Q3: Line 106. The following article compares VGG and ResNet architectures and can be added here: https://doi.org/10.3390/s21238083

A3: The reference has been added in Section 2.1.

Q4: Also, some other recently developed visual transformer methods with applications can be added in Related Works, such as https://doi.org/10.3390/rs14061400

A4: The reference has been added in Section 2.1.

Q5: Line 194. Format issue. It should be “Figure 4”. Please check and correct similar issues. Like line 227 that should say “Table 1”. Likewise, when you referring to equation, you should say “equation (x)”.

A5: We have checked the paper and corrected them.

Q6: Please add the computational time (in seconds or big O) for different methods in one of the tables.

A6: The complexity comparison has been added. The computational time of different methods is difficult to count since the training/validation procedure on such “big” datasets is quite “expensive”. The FLOPs and Params can be used as reference values to evaluate the calculation time.

Q7: The discussion section is very short. Please first state the objective and discuss your results in light of other similar studies. Furthermore, please include the limitations of the study and give future directions.

A7: We have extended the discussion based on the suggestions.

Q8: Finally, please carefully proofread the article and fix typos/punctuation/grammar issues.

A8: We have carefully checked our paper and improved the writing.

Round 2

Reviewer 3 Report

Authors  revised their  manuscript taking into account all my suggestions. I consider this paper publishable in the present form.